# Confidence-based Ensembling of Perspective-aware Models

**Silvia Casola**[1], **Soda Marem Lo**[1], **Valerio Basile**[1], **Simona Frenda**[1,2]
**Alessandra Teresa Cignarella**[1,2], **Viviana Patti**[1] and **Cristina Bosco**[1]
[1]University of Turin, Italy
[2]aequa-tech, Turin, Italy

## Abstract

Research in the field of NLP has recently focused on the variability that people show in selecting labels when performing an annotation task. Exploiting disagreements in annotations has been shown to offer advantages for accurate modelling and fair evaluation. In this paper, we propose a strongly perspectivist model for supervised classification of natural language utterances. Our approach combines the predictions of several perspective-aware models using key information of their individual confidence to capture the subjectivity encoded in the annotation of linguistic phenomena. We validate our method through experiments on two case studies, irony and hate speech detection, in in-domain and cross-domain settings. The results show that confidence-based ensembling of perspective-aware models seems beneficial for classification performance in all scenarios. In addition, we demonstrate the effectiveness of our method with automatically extracted perspectives from annotations when the annotators' metadata are not available.

## 1 Introduction

Human label variability (Plank, 2022), traditionally considered a source of noise in annotated data, has recently become the subject of a series of research works in Natural Language Processing. Leveraging disagreement in annotation has been found beneficial towards more accurate modelling of natural language phenomena (Uma et al., 2020a) and fairer evaluation (Basile et al., 2021).

These observations are particularly relevant when the focus of the study is on some latent linguistic phenomena at the level of pragmatics, such as irony or hateful speech, because of the high degree of subjectivity involved in the annotation of these phenomena (Basile, 2020). Taking this research direction one step further, the *perspectivist* approach (Cabitza et al., 2023) aims at creating models capable of encoding different human points of view on observed phenomena (Abercrombie et al., 2022; Mastromattei et al., 2022).

This paper proposes a strongly perspectivist model for the supervised classification of natural language utterances. While our method is agnostic to the specific phenomenon that is the subject of classification, we expect it to work well when the focus is on pragmatic language phenomena. The reason for this expectation lies in the subjectivity of the annotation process for this level of analysis, which our model aims to model and exploit for a better classification.

To prove it, we tested this methodology in two largely explored pragmatic phenomena in NLP: irony and hate speech detection. Irony is a figurative language device that conveys the opposite or an extension of the literal meaning; hate speech produces the incitement of hate and violence against a specific target (individuals or entire communities) or the reinforcement of negative stereotypes (Sanguinetti et al., 2018). All experiments are on datasets in English.

To test the robustness of the proposed perspectivist method, we performed in-domain and cross-domain experiments, i.e., testing models fine-tuned on a dataset on data from the same set and from different sets.

We took inspiration from the observations in (Frenda et al., 2023b) about the higher confidence exhibited by perspective-aware models trained on longitudinal subsets of the disaggregated data with respect to models trained on aggregated data. Starting from this observation, we propose a method to ensemble perspective-aware models that take model confidence into account.

Furthermore, we prove that our methodology can also be applied when the annotators' metadata are unavailable and describe a methodology to mine perspectives from annotations directly. With this approach, on the one hand, we test whether our method can be applied when no explicit demo-

graphic data of the annotators are available. On the other hand, we test whether the annotations can be used to identify interesting implicit discriminative dimensions that differ from those encoded in the metadata.

This paper is organized as follows. In Section 2, we present related previous works, with a specific focus on methods to build perspective-aware models; Section 3 describes our perspectivist method. We perform in-domain and cross-domain experiments for irony (Section 4) and hate speech detection (Section 5). We detail our approach to mine perspectives and the results obtained with these implicit perspectives in Section 6. The discussion and conclusive observations that emerged from our work are in Section 7.

## 2 Related work

The development of computational methodologies to account for the differing viewpoints of different annotators in detecting pragmatic phenomena is not new. However, most previous research focused on taking advantage of disagreement in the data to improve the classification performance (Aroyo and Welty, 2015; Plank et al., 2014; Jamison and Gurevych, 2015).

Recently, attention to the disagreement among the annotators has been increasing. Uma et al. (2021) and Leonardellli et al. (2023) organized two editions of the LeWiDi (Learning with disagreement) shared tasks, respectively, in SemEval 2021 (Task 12) and SemEval 2023 (Task 11). The organizers of both editions of the workshop remark on the concept of *soft label* (vs. *hard label*) proposed by Uma et al. (2020b), a method for learning from a distribution over the label space to maximize classification performance and provide a robust evaluation metric. The paradigm of *learning from disagreement* takes into account the presence of disagreement in the annotated data, because different opinions could arise from different annotators.

Data *perspectivism* overcomes the idea of "ground truth" in constructing datasets and creating NLP models to give space to different perspectives that can be encoded in the annotated corpora, focusing on *who the annotators are*. Following this new theoretical framework, two editions of the NLPerspectives workshop[1] have been organized in 2022 and 2023.

[1] https://nlperspectives.di.unito.it/w/

For modelling perspectives in hate speech detection, scholars have proposed various approaches, taking into account the individual annotation of each annotator (Davani et al., 2022), group-based annotations (Akhtar et al., 2020), specific aspects of annotators such as biases (Milkowski et al., 2021) and beliefs (Kocoń et al., 2021a), or using personalized methods (Kocoń et al., 2021b).

In particular, Davani et al. (2022) proposed to exploit annotators' systematic disagreement and investigated multiple approaches for hate speech and emotions detection: an ensemble model, where each classifier is trained on annotator-specific data only; a multi-label model, where each label corresponds to that assigned by an individual annotator; a multi-task model, where the prediction of labels from each annotator is modelled as a separate task. The latter is superior to a baseline model trained on the aggregated data. Akhtar et al. (2020), instead, divided the annotators into two groups; they took into account common personal characteristics such as ethnicity, social background, and culture and modelled the group-based perspectives in recognition of hate speech.

In contrast to *perspectivist* literature on hate speech detection, perspectivist approaches to irony detection are less explored. To our knowledge, only two disaggregated datasets for English exist on humour (Simpson et al., 2019) and irony (Frenda et al., 2023b). The first dataset was used as a benchmark in the first edition of LeWiDi (Uma et al., 2021). The second, the English Perspectivist Irony Corpus (EPIC), was used to explore the agreement among annotators with similar socio-demographic traits. Based on these traits, the authors also created perspectivist models showing the variation of the perception of irony among different population segments.

## 3 Confidence-based perspectivist ensemble

Inspired by studies that demonstrate how the perception of irony and toxicity varies based on sociodemographic identities (Sap et al., 2022; Frenda et al., 2023a), we divided annotators based on their metadata (gender, age, nationality) and constructed perspective-specific datasets encoding their perception of a given phenomenon (see details in Section 4.1). For each of these longitudinal splits of the training set, we fine-tuned a pre-trained language model (PLM), obtaining 11 classification models

for irony detection. While these models are trained on mostly similar texts, we expect differences to emerge in their predictions due to the different labels they were exposed to at training time.

Furthermore, the models will output predictions with different degrees of confidence, which is the critical information we exploit for the proposed ensemble model. Formally, for each instance at inference time, we obtain from the trained models $n$ predictions $(l_1, c_1), ..., (l_n, c_n)$ where $l_i \in \{0, 1\}$ is a binary label and $c_i \in [0, 1]$ is a confidence score. We use the normalized difference between the two logits rescaled by softmax (Taha et al., 2022) as a confidence score. Specifically, we compute the confidence as:

$$c_k = \begin{cases} \frac{L_1 - L_0}{|L_1 + L_0|} & \text{if } l_k = 1 \\ \frac{L_0 - L_1}{|L_1 + L_0|} & \text{if } l_k = 0 \end{cases}$$

where $L_c$ is the logit of class $c$ rescaled by softmax and $l_k$ is the model prediction.

The predicted label of our confidence-based ensemble model is therefore computed as:

$$l_{weight} = argmax \left( \sum_{l_k=0} c_k, \sum_{l_k=1} c_k \right)$$

In plain terms, the confidence scores of the models predicting the negative class are summed, as are the confidence scores of the models predicting the positive class, and the ensemble selects the label associated with the highest confidence sum.

We also propose a simpler version of the ensemble computing a confidence-based vote:

$$l_{high} = l_k : k = argmax(c_1, ..., c_n)$$

In this version, the ensemble selects the label predicted by the classifier with the highest confidence in its prediction.

## 4 Perspectivist Classification of Irony

We present an irony detection experiment with the goal of proving the informative value of annotator perspectives in the annotated data.

### 4.1 Data

The dataset used for testing our methodology in irony detection is EPIC (English Perspectivist Irony Corpus) released by Frenda et al. (2023b). EPIC contains 3,000 short conversations (post and reply) from Twitter and Reddit annotated by 74 annotators coming from 5 English-speaking countries (United States, United Kingdom, Australia, Ireland, and India) and reporting different socio-demographic traits. We used the data split proposed in the original publication for our experiments. Starting from the entire set of EPIC data, we computed the majority voting among the annotations of each instance, creating an aggregated set of 2,767 instances[2]. We then split the data into training (80%) and test set (20%), reproducing the same distribution of tweets and comments from Reddit. The aggregated training set was used to train the *non-perspectivist models* (NP) (see Section 4.3), while the test set was used to test all the models in the in-domain setting (Table 1). After setting apart the test instances, we reproduced the longitudinal split from the EPIC paper based on the annotators' socio-demographic traits, resulting in 11 sub-sets of different sizes, each representing an annotator perspective: Female (1,952), Male (2,023), Boomer (441), GenX (1,757), GenY (1,964), GenZ (1,124), UK (1,365), India (1,175), Irish (1,296), US (1,352), Australian (1,377).

The details are reported in Appendix A (Table 8).

### 4.2 Experimental Setting

We fine-tuned three general-purpose PLMs. We aimed to test the method with well-established baselines, and therefore selected two of the most widely used PLMs, namely BERT (Devlin et al., 2019) and RoBERTa (Zhuang et al., 2021). Since our method implies multiple fine tunings with potential ecological and computational impact, we further checked its validity on a smaller model i.e., DistilBERT (Sanh et al., 2019). We downloaded pre-trained models from the Huggingface repository with identifiers *bert-base-uncased*, *roberta-base* and *distilbert-base-uncased* respectively. We used a batch size of 16 and a learning rate of $5e - 5$ for BERT and $5e - 6$ for ROBERTA and DISTIL-BERT. We customized the model to implement Focal Loss (Lin et al., 2017) to prevent overfitting in the case of unbalanced datasets. We implemented early stopping, with patience of 2 epochs on the validation loss for all the models.

### 4.3 Baselines

We experimentally compared our model with several baselines and other approaches to determine

---

[2]Instances were removed when no majority was reached.

| PLM | model | negative class | | | positive class | | | macro-average | | | Acc. |
|---|---|---|---|---|---|---|---|---|---|---|---|
| | | prec. | rec. | F1 | prec. | rec. | F1 | prec. | rec. | F1 | |
| BERT | NP | .873 | .711 | .780 | .342 | .581 | .425 | .608 | .646 | $.603_{\pm.036}$ | $.685_{\pm.059}$ |
| | I-ENS$_{high}$ | .880 | .701 | .780 | .339 | .614 | .436 | .610 | .657 | $.608_{\pm.018}$ | $.684_{\pm.026}$ |
| | I-ENS$_{weight}$ | .873 | .737 | .799 | .350 | .568 | .432 | .611 | .652 | $.615_{\pm.016}$ | $.703_{\pm.020}$ |
| | A-ENS$_{high}$ | .897 | .590 | .711 | .307 | .728 | .431 | .602 | .659 | $.571_{\pm.021}$ | $.618_{\pm.028}$ |
| | A-ENS$_{weight}$ | .897 | .596 | .715 | .307 | .722 | .431 | .602 | .659 | $.573_{\pm.013}$ | $.621_{\pm.019}$ |
| | M-ENS | .868 | .745 | .801 | .349 | .543 | .423 | .608 | .644 | $.612_{\pm.016}$ | $.705_{\pm.026}$ |
| | *C-ENS$_{high}$* | .876 | .701 | .779 | .335 | .603 | .430 | .606 | .652 | $.605_{\pm.014}$ | $.682_{\pm.018}$ |
| | *C-ENS$_{weight}$* | .875 | .743 | .803 | .358 | .571 | .439 | .616 | .657 | $\mathbf{.621}_{\pm.021}$ | $\mathbf{.709}_{\pm.026}$ |
| DISTILBERT | NP | .894 | .658 | .757 | .332 | .685 | .447 | .613 | .671 | $.602_{\pm.011}$ | $.663_{\pm.015}$ |
| | I-ENS$_{high}$ | .873 | .689 | .770 | .324 | .597 | .420 | .598 | .643 | $.595_{\pm.015}$ | $.671_{\pm.016}$ |
| | I-ENS$_{weight}$ | .889 | .645 | .748 | .321 | .676 | .436 | .605 | .661 | $.592_{\pm.012}$ | $.652_{\pm.014}$ |
| | A-ENS$_{high}$ | .891 | .645 | .748 | .323 | .683 | .439 | .607 | .664 | $.593_{\pm.010}$ | $.652_{\pm.012}$ |
| | A-ENS$_{weight}$ | .889 | .645 | .747 | .321 | .676 | .435 | .605 | .660 | $.591_{\pm.007}$ | $.651_{\pm.010}$ |
| | M-ENS | .877 | .712 | .786 | .341 | .600 | .435 | .609 | .656 | $.610_{\pm.011}$ | $.690_{\pm.15}$ |
| | *C-ENS$_{high}$* | .879 | .712 | .786 | .343 | .605 | .438 | .611 | .658 | $\mathbf{.612}_{\pm.013}$ | $.690_{\pm.017}$ |
| | *C-ENS$_{weight}$* | .878 | .713 | .787 | .344 | .603 | .438 | .611 | .658 | $\mathbf{.612}_{\pm.014}$ | $\mathbf{.691}_{\pm.018}$ |
| ROBERTA | NP | .916 | .702 | .793 | .386 | .740 | .506 | .651 | .721 | $.649_{\pm.030}$ | $.710_{\pm.042}$ |
| | I-ENS$_{high}$ | .898 | .736 | .809 | .384 | .664 | .487 | .641 | .700 | $.648_{\pm.010}$ | $.721_{\pm.010}$ |
| | I-ENS$_{weight}$ | .901 | .723 | .802 | .379 | .679 | .486 | .640 | .701 | $.644_{\pm.006}$ | $.714_{\pm.009}$ |
| | A-ENS$_{high}$ | .912 | .655 | .762 | .350 | .747 | .476 | .631 | .701 | $.619_{\pm.009}$ | $.673_{\pm.014}$ |
| | A-ENS$_{weight}$ | .913 | .648 | .758 | .347 | .752 | .475 | .630 | .700 | $.616_{\pm.015}$ | $.669_{\pm.019}$ |
| | M-ENS | .897 | .760 | .823 | .403 | .649 | .496 | .650 | .704 | $.659_{\pm.010}$ | $.738_{\pm.014}$ |
| | *C-ENS$_{high}$* | .904 | .748 | .818 | .401 | .680 | .505 | .653 | .714 | $.661_{\pm.009}$ | $.734_{\pm.009}$ |
| | *C-ENS$_{weight}$* | .901 | .758 | .823 | .406 | .667 | .505 | .654 | .712 | $\mathbf{.664}_{\pm.011}$ | $\mathbf{.739}_{\pm.012}$ |

Table 1: In-domain performance, i.e., models trained on EPIC and tested on the EPIC gold test set. The proposed confidence-based models (C-ENS) are marked in italics. For the macro-average F1 score and the accuracy score, we also report the standard deviation.

| PLM | model | negative class | | | positive class | | | macro-average | | | Acc. |
|---|---|---|---|---|---|---|---|---|---|---|---|
| | | prec. | rec. | F1 | prec. | rec. | F1 | prec. | rec. | F1 | |
| BERT | NP | .592 | .641 | .601 | .340 | .315 | .303 | .466 | .478 | $.452_{\pm.041}$ | $.511_{\pm.039}$ |
| | I-ENS$_{high}$ | .564 | .611 | .580 | .297 | .280 | .279 | .431 | .445 | $.430_{\pm.030}$ | $.480_{\pm.015}$ |
| | I-ENS$_{weight}$ | .575 | .599 | .575 | .315 | .317 | .302 | .445 | .458 | $.438_{\pm.029}$ | $.487_{\pm.014}$ |
| | A-ENS$_{high}$ | .612 | .445 | .506 | .396 | .562 | .460 | .504 | .503 | $\mathbf{.483}_{\pm.023}$ | $.491_{\pm.025}$ |
| | A-ENS$_{weight}$ | .616 | .399 | .474 | .394 | .608 | .473 | .505 | .504 | $.473_{\pm.016}$ | $.482_{\pm.016}$ |
| | M-ENS | .595 | .572 | .577 | .364 | .399 | .373 | .480 | .486 | $.475_{\pm.039}$ | $.504_{\pm.016}$ |
| | *C-ENS$_{high}$* | .602 | .589 | .587 | .377 | .397 | .376 | .490 | .493 | $.481_{\pm.031}$ | $\mathbf{.513}_{\pm.018}$ |
| | *C-ENS$_{weight}$* | .596 | .598 | .590 | .365 | .375 | .360 | .481 | .486 | $.475_{\pm.038}$ | $.509_{\pm.018}$ |
| DISTILBERT | NP | .625 | .460 | .528 | .411 | .576 | .478 | .518 | .518 | $\mathbf{.503}_{\pm.007}$ | $.506_{\pm.009}$ |
| | I-ENS$_{high}$ | .588 | .683 | .631 | .350 | .269 | .302 | .469 | .476 | $.466_{\pm.034}$ | $.519_{\pm.017}$ |
| | I-ENS$_{weight}$ | .597 | .673 | .632 | .372 | .305 | .332 | .485 | .489 | $.482_{\pm.034}$ | $.527_{\pm.018}$ |
| | A-ENS$_{high}$ | .604 | .643 | .621 | .384 | .351 | .364 | .494 | .497 | $.492_{\pm.038}$ | $.527_{\pm.022}$ |
| | A-ENS$_{weight}$ | .601 | .641 | .619 | .383 | .347 | .361 | .492 | .494 | $.490_{\pm.035}$ | $.525_{\pm.021}$ |
| | M-ENS | .601 | .726 | .656 | .381 | .265 | .308 | .491 | .496 | $.482_{\pm.033}$ | $.543_{\pm.013}$ |
| | *C-ENS$_{high}$* | .612 | .676 | .64 | .399 | .342 | .363 | .506 | .509 | $.501_{\pm.038}$ | $.543_{\pm.013}$ |
| | *C-ENS$_{weight}$* | .608 | .708 | .652 | .398 | .301 | .338 | .503 | .504 | $.495_{\pm.028}$ | $\mathbf{.547}_{\pm.012}$ |
| ROBERTA | NP | .717 | .524 | .598 | .486 | .679 | .563 | .602 | .601 | $.581_{\pm.026}$ | $.585_{\pm.026}$ |
| | I-ENS$_{high}$ | .693 | .542 | .605 | .476 | .629 | .540 | .584 | .586 | $.572_{\pm.019}$ | $.577_{\pm.020}$ |
| | I-ENS$_{weight}$ | .706 | .516 | .592 | .476 | .667 | .554 | .591 | .592 | $.573_{\pm.024}$ | $.576_{\pm.023}$ |
| | A-ENS$_{high}$ | .750 | .496 | .591 | .492 | .739 | .588 | .621 | .618 | $.590_{\pm.025}$ | $.593_{\pm.025}$ |
| | A-ENS$_{weight}$ | .765 | .471 | .578 | .492 | .773 | .599 | .629 | .622 | $.589_{\pm.027}$ | $.591_{\pm.026}$ |
| | M-ENS | .699 | .645 | .668 | .514 | .570 | .536 | .606 | .608 | $.602_{\pm.019}$ | $.615_{\pm.010}$ |
| | *C-ENS$_{high}$* | .711 | .620 | .661 | .515 | .615 | .560 | .613 | .617 | $\mathbf{.610}_{\pm.015}$ | $.618_{\pm.014}$ |
| | *C-ENS$_{weight}$* | .708 | .630 | .665 | .517 | .601 | .555 | .612 | .616 | $\mathbf{.610}_{\pm.011}$ | $\mathbf{.619}_{\pm.010}$ |

Table 2: Cross-domain performance, i.e., models trained on EPIC and tested on the SemEval 2018 Task 3 test set.

the source of the improvement.

**Non-perspectivist model (NP)** For each data instance, we aggregated all annotators' labels through majority voting. We discarded utterances for which no majority was found, as discussed in Section 4.1. We then used this dataset to fine-tune the three PLMs. This approach does not exploit the annotators' disagreement.

**Instance-based Ensemble (I-ENS)** This baseline is an ensemble of non-perspectivist models.

Each model was trained on a sample of the aggregated dataset, which had the same cardinality as the perspective-aware ones. We predicted the final label using the two confidence-based approaches discussed in Section 3.

**Annotator-based Ensemble (A-ENS)** We wanted to quantify whether grouping annotators by their demographic characteristics was advantageous over using random groups. To this end, we randomly selected annotators from the entire pool for each model in the ensemble. We then created 11 sub-datasets based only on the majority voting of such annotators, which we used to train the models; we predicted the final label using our confidence-based methods. The number of annotators whose labels are used in each sub-dataset is consistent with the number of annotators in each perspectivist dataset.

**Majority voting Ensemble (M-ENS)** We wanted to test the contribution of using confidence information as an ensembling strategy. To this end, we compared our model to an identical ensemble that computes the final label using a simple majority voting strategy over the output of each classifier.

For all baselines, the three PLMs were trained with the same hyperparameters and training strategies.

### 4.4 Results

Table 1 shows the experiment results, reporting an average of 10 runs. With all three PLMs, the performance of the classification provided by our ensemble models on the gold standard test set overcome the baselines and are always better than all the other models. C-ENS$_{weight}$, which computes its prediction based on the weighted sum of each model's label, achieves a better result, in terms of f1 score, over C-ENS$_{high}$, which selects the model with the highest confidence in the prediction. However, the latter is still superior to most baseline results.

For all models, we notice a substantial improvement when annotators are grouped by their characteristics rather than randomly, which validates our original hypothesis about the informativeness of perspectivist data. Most importantly, using a confidence measure consistently leads to better results than simply ensembling by voting. This confirms confidence as a key piece of information for perspectivist modelling.

### 4.5 Cross-domain Classification of Irony

To test the robustness of our method, we tested our ensemble model in a cross-domain setting. Given the models trained on the EPIC dataset that we described in Section 4, we tested them on a different irony detection dataset, with no particular adaptation. In particular, we use the test set from SemEval 2018 Task 3: "Irony Detection in English Tweets" (Van Hee et al., 2018).

With respect to the in-domain performance, the results of the cross-domain experiment are more mixed and less straightforward, as shown in Table 2. Nevertheless, the general trend confirms the advantage of using disaggregated data, perspectives, and model confidence, with notable exceptions: A-ENS$_{high}$ with BERT and the non-perspectivist baseline (NP) with DISTILBERT, in particular, gave particularly high results on this task. In both cases, the results are mostly due to better performance on the positive class (irony), indicating the need to better control model bias during the training of perspective-aware models. However, the C-ENS classifiers always rank in the top-three results, and the best result by far is obtained with ROBERTA paired with either C-ENS models.

## 5 Perspectivist Hate Speech Detection

Considering the promising results achieved by our method on the irony detection task presented in the previous sections, we further validated our approach on a different task. Hate speech (HS), similarly to irony, is a pragmatic phenomenon in language, with a high degree of subjectivity involved in its annotation (Poletto et al., 2021). Unsurprisingly, HS and related phenomena of undesirable language are objects of several works in perspectivist NLP (Cercas Curry et al., 2021; Akhtar et al., 2021; Dinu et al., 2021). This section presents the results of experiments analogous to the one we presented on irony detection on similar-shaped data annotated with HS.

### 5.1 Data

The dataset used in this set of experiments is the Measuring Hate Speech corpus (MHS) (Kennedy et al., 2020)[3], which contains identity group targets and the annotators' sociodemographic information, as well as their estimated survey interpretation bias, difficulty, and rarity of decision. MHS contains

---

[3] https://huggingface.co/datasets/ucberkeley-dlab/measuring-hate-speech

| PLM | model | negative class | | | positive class | | | macro-average | | | Acc. |
|---|---|---|---|---|---|---|---|---|---|---|---|
| | | prec. | rec. | F1 | prec. | rec. | F1 | prec. | rec. | F1 | |
| BERT | NP | .898 | .818 | .856 | .680 | .806 | .737 | .789 | .812 | .796$_{\pm.008}$ | .814$_{\pm.010}$ |
| | M-ENS | .912 | .804 | .854 | .671 | .837 | .745 | .792 | .820 | .800$_{\pm.003}$ | .814$_{\pm.003}$ |
| | *C-ENS$_{high}$* | .912 | .800 | .852 | .668 | .838 | .743 | .790 | .819 | .798$_{\pm.004}$ | .813$_{\pm.005}$ |
| | *C-ENS$_{weight}$* | .913 | .805 | .856 | .674 | .840 | .748 | .794 | .823 | **.802**$_{\pm.003}$ | **.816**$_{\pm.004}$ |
| DISTILBERT | NP | .906 | .815 | .858 | .680 | .822 | .745 | .793 | .819 | **.801**$_{\pm.003}$ | **.817**$_{\pm.004}$ |
| | M-ENS | .913 | .801 | .853 | .669 | .841 | .745 | .791 | .821 | .799$_{\pm.002}$ | .814$_{\pm.002}$ |
| | *C-ENS$_{high}$* | .911 | .800 | .852 | .667 | .837 | .742 | .789 | .818 | .797$_{\pm.003}$ | .812$_{\pm.003}$ |
| | *C-ENS$_{weight}$* | .912 | .801 | .853 | .669 | .839 | .744 | .791 | .820 | .799$_{\pm.001}$ | .813$_{\pm.002}$ |
| ROBERTA | NP | .908 | .802 | .852 | .668 | .830 | .740 | .788 | .816 | .796$_{\pm.005}$ | .811$_{\pm.006}$ |
| | M-ENS | .911 | .805 | .855 | .672 | .836 | .745 | .792 | .820 | .800$_{\pm.002}$ | .815$_{\pm.003}$ |
| | *C-ENS$_{high}$* | .912 | .807 | .856 | .675 | .837 | .747 | .793 | .822 | **.802**$_{\pm.003}$ | **.817**$_{\pm.003}$ |
| | *C-ENS$_{weight}$* | .911 | .808 | .856 | .676 | .835 | .747 | .793 | .822 | **.802**$_{\pm.001}$ | **.817**$_{\pm.002}$ |

Table 3: In-domain performance of Hate Speech detection models, i.e., models trained on MHS and tested on the MHS gold test set. The proposed confidence-based models (C-ENS) are marked in italics. For the macro-average F1 score and the accuracy score, we also report the standard deviation.

| PLM | model | negative class | | | positive class | | | macro-average | | | Acc. |
|---|---|---|---|---|---|---|---|---|---|---|---|
| | | prec. | rec. | F1 | prec. | rec. | F1 | prec. | rec. | F1 | |
| BERT | NP | .602 | .468 | .526 | .441 | .576 | .500 | .522 | .522 | .513$_{\pm.014}$ | .514$_{\pm.014}$ |
| | M-ENS | .615 | .472 | .534 | .451 | .595 | .513 | .533 | .533 | .523$_{\pm.003}$ | **.524**$_{\pm.003}$ |
| | *C-ENS$_{high}$* | .617 | .470 | .533 | .452 | .600 | .515 | .534 | .535 | **.524**$_{\pm.010}$ | **.524**$_{\pm.010}$ |
| | *C-ENS$_{weight}$* | .613 | .471 | .533 | .449 | .592 | .511 | .531 | .532 | .522$_{\pm.004}$ | .522$_{\pm.004}$ |
| DISTILBERT | NP | .620 | .493 | .550 | .457 | .586 | .513 | .539 | .539 | .531$_{\pm.002}$ | .532$_{\pm.003}$ |
| | M-ENS | .629 | .502 | .558 | .465 | .594 | .522 | .547 | .548 | .540$_{\pm.003}$ | .541$_{\pm.004}$ |
| | *C-ENS$_{high}$* | .631 | .515 | .567 | .469 | .587 | .521 | .550 | .551 | **.544**$_{\pm.003}$ | **.545**$_{\pm.003}$ |
| | *C-ENS$_{weight}$* | .629 | .507 | .562 | .466 | .590 | .521 | .548 | .549 | .541$_{\pm.003}$ | .542$_{\pm.004}$ |
| ROBERTA | NP | .610 | .488 | .542 | .448 | .571 | .502 | .529 | .530 | .522$_{\pm.005}$ | .523$_{\pm.005}$ |
| | M-ENS | .613 | .495 | .548 | .452 | .571 | .505 | .533 | .533 | **.526**$_{\pm.005}$ | .527$_{\pm.005}$ |
| | *C-ENS$_{high}$* | .613 | .497 | .549 | .452 | .569 | .504 | .532 | .533 | **.526**$_{\pm.006}$ | .527$_{\pm.007}$ |
| | *C-ENS$_{weight}$* | .613 | .497 | .549 | .452 | .569 | .504 | .533 | .533 | **.526**$_{\pm.005}$ | **.528**$_{\pm.005}$ |

Table 4: Cross-domain performance of Hate Speech detection models, i.e., models trained on MHS and tested on the HatEval test set.

39,565 texts annotated by 7,912 people. The HS label in MHS has three values (no HS, weak, and strong HS) that we combine into a binary label conflating weak and strong HS. From the available metadata, we selected 4 dimensions (education, gender, ideology, and income, see details in Table 9 in Appendix A) and aggregated the annotators by their possible values, and split the dataset randomly across the comments into training (80%), validation (10%), and test (10%) sets.

## 5.2 Results

Table 3 illustrates the results when testing the models in-domain, showing that the confidence-based models perform better than the baselines with ROBERTA, while the results are mixed with the other two PLMs. Given the dataset size — which is an order of magnitude larger than EPIC — we could not include the ensemble-based baselines; adding the baseline results would have required fine-tuning 9 PLMs per run (with a grand total of 540 models), which would be unfeasible given

our computational resources, and possibly unnecessarily impactful on the environment. Comparing the two ensemble models, C-ENS$_{weight}$ tends to give higher scores, except for the ROBERTA PLM, where the two have the same performance.

## 5.3 Cross-domain Classification of Hate Speech

We further tested our models on the test set of the English part of HatEval (Basile et al., 2019), a popular benchmark for HS detection from the SemEval 2019 Task 5: "Multilingual Detection of Hate Speech Against Immigrants and Women in Twitter." The HatEval dataset consists of 9,000 instances for the training set, 1,000 for the validation set, and 2,972 for the test set.

The results in Table 4 show how the confidence-based models give the best results in cross-domain HS detection with all three PLMs. Differently from the in-domain performance, in this case, the *C-ENS$_{high}$* tends to outperform the *C-ENS$_{weight}$*. We attribute this trend to the difference in the annota-

tors' backgrounds from training to test set in the cross-domain setting. Since the socio-demographic makeup of the annotators of HatEval is different from EPIC, the $C - ENS_{high}$ ensemble it is likely able to pick up on their intersection, while in $C - ENS_{weight}$, the opinion of a broader set of annotators (possibly not intersecting with HatEval) has a stronger influence on the final prediction.

## 6 Mining perspectives

The method we presented so far takes advantage of some division of the annotators known in advance. However, this is not always the case, for instance, when an annotation results from crowd-sourcing with anonymous contributors. Therefore, we devised a method to automatically group the annotators meaningfully to represent different perspectives. In this section, we present the result of a similar experiment to the main one presented in Section 4, with the difference that the perspectives are mined automatically. In this setting, the data split to train the perspective-aware models depends on annotations rather than demographic traits.

Following Fell et al. (2021), given $n$ annotators and $k$ instances, we constructed a label matrix $V^{n \times k}$. Since each instance was labeled, on average, by $4.72$ annotators only, the resulting matrix was highly sparse. We modelled the similarity among annotators in terms of inter-annotator agreement, using Krippendorff's alpha ($\alpha$) (Krippendorff, 2011). We thus computed the pairwise agreement between annotators $i, j$, obtaining a similarity matrix $A_{i,j} = \alpha(V_{i:}, V_{j:})$, from which we calculated the distance matrix $D = 1 - A$, used as input for the clustering algorithm.

We found $82$ cases in which two annotators had no common annotated instances, and we decided to assign $\alpha = 0$ for all of them. In $158$ cases, there was perfect agreement between pairs of annotators on a small sample of data, having as a consequence that the $\alpha$ was not well-defined — Checco et al. (2017) describes it as a "paradox". We set $\alpha = 1$ for these pairs. We used hierarchical agglomerative clustering[4] with Ward's linkage criterion to group the annotators, computing the full dendrogram. Then, we applied the Calinski-Harabasz (Caliński and Harabasz, 1974) and Davies-Bouldin Indexes (Davies and Bouldin, 1979) to assess the best number of clusters between 2 and 5, and 11 clusters as the perspectives. The two metrics re-

[4]From Scikit Learn

| cluster | 0 | 1 | 2 | 3 | 4 |
|---|---|---|---|---|---|
| Node | 6.307 | 5.919 | 6.639 | 5.600 | 5.444 |
| Annotators | 18 | 12 | 19 | 15 | 10 |
| Female | 34% | 11% | 31% | 9% | 14% |
| Male | 15% | 21% | 21% | 31% | 13% |
| Australian | 27% | 13% | 20% | 27% | 13% |
| Indian | 33% | 33% | 20% | 7% | 7% |
| Irish | 27% | 20% | 33% | 7% | 13% |
| British | 20% | 0% | 27% | 40% | 13% |
| US american | 14% | 14% | 29% | 21% | 21% |
| Boomer | 0% | 0% | 33% | 33% | 33% |
| GenX | 36% | 18% | 14% | 14% | 18% |
| GenY | 18% | 16% | 32% | 21% | 13% |
| GenZ | 30% | 10% | 30% | 30% | 0% |

Table 5: Description of the clusters in respect to the node level at which they joined; the number of annotators; the distribution of the demographic traits among clusters.

spectively measure how dense and how similar the clusters are; thus, we opted for a trade-off between the two, minimizing their ratio, and resulting in 5 clusters described in Table 5.

Looking at the annotators' distribution, there is no evident mapping between the demographic-based perspectives and the mined clusters, for any of the gender, nationality, and generation dimensions. This confirms our hypothesis that the metadata provided by EPIC do not necessarily induce the best possible split among the annotators, and therefore automatically mining perspectives could uncover latent dimension that better discriminates the annotators. As for gender, cluster 3 represents a small percentage of females, but male annotators are better distributed among the five clusters. Unbalanced representations can be seen also when considering nationality; cluster 3 hosts 40% of the British annotators, and a small percentage of both Indian and Irish annotators (represented especially in the first three clusters), while cluster 1 has no British annotators at all. Similarly, focusing on age, cluster 0 and 1 have no Boomer annotators, and GenZ is more represented by clusters 0, 2 cluster 3 in respect to the remaining two. Despite these differences, none of the demographic groups merge homogeneously into specific clusters.

Finally, we tested the validity of the clusters by reproducing the method described in Section 3. We obtained $5$ classification models, which we tested in an in-domain setting (Section 4) and in a cross-domain one (Section 4.5); in both cases, we averaged the results over 10 runs. Table 6 shows that the macro-average F1 scores are always

| PLM | model | negative class | | | positive class | | | macro-average | | | Acc. |
|---|---|---|---|---|---|---|---|---|---|---|---|
| | | prec. | rec. | F1 | prec. | rec. | F1 | prec. | rec. | F1 | |
| BERT | $C\text{-}ENS_{high}$ | .887 | .679 | .768 | .338 | .651 | .443 | .613 | .665 | $.605_{\pm.024}$ ($\Delta + .000$) | $.673_{\pm.038}$ |
| | $C\text{-}ENS_{weigh}$ | .887 | .709 | .787 | .354 | .634 | .452 | .620 | .671 | $.620_{\pm.017}$ ($\Delta - .001$) | $.694_{\pm.028}$ |
| DISTILBERT | $C\text{-}ENS_{high}$ | .877 | .725 | .794 | .348 | .590 | .437 | .612 | .657 | $.616_{\pm.009}$ ($\Delta + .004$) | $.698_{\pm.012}$ |
| | $C\text{-}ENS_{weigh}$ | .877 | .727 | .795 | .350 | .589 | .438 | .613 | .658 | $.617_{\pm.011}$ ($\Delta + .005$) | $.700_{\pm.015}$ |
| ROBERTA | $C\text{-}ENS_{high}$ | .907 | .736 | .812 | .396 | .695 | .504 | .651 | .716 | $.658_{\pm.014}$ ($\Delta - .003$) | $.728_{\pm.016}$ |
| | $C\text{-}ENS_{weigh}$ | .907 | .753 | .823 | .410 | .689 | .514 | .658 | .721 | $.668_{\pm.015}$ ($\Delta + .004$) | $.740_{\pm.016}$ |

Table 6: In-domain performance of cluster-based models, trained on EPIC and tested on the EPIC gold test set. In brackets the delta between the cluster-based and the correspondent demographic-based ensemble model.

| PLM | model | negative class | | | positive class | | | macro-average | | | Acc. |
|---|---|---|---|---|---|---|---|---|---|---|---|
| | | prec. | rec. | F1 | prec. | rec. | F1 | prec. | rec. | F1 | |
| BERT | $C\text{-}ENS_{high}$ | .629 | .404 | .467 | .400 | .613 | .470 | .515 | .509 | $.469_{\pm.037}$ ($\Delta - .012$) | $.487_{\pm.045}$ |
| | $C\text{-}ENS_{weigh}$ | .602 | .438 | .489 | .380 | .552 | .437 | .491 | .495 | $.463_{\pm.037}$ ($\Delta - .012$) | $.483_{\pm.037}$ |
| DISTILBERT | $C\text{-}ENS_{high}$ | .601 | .694 | .642 | .379 | .295 | .325 | .490 | .495 | $.484_{\pm.032}$ ($\Delta - .017$) | $.536_{\pm.012}$ |
| | $C\text{-}ENS_{weigh}$ | .597 | .730 | .655 | .372 | .249 | .294 | .484 | .489 | $.474_{\pm.026}$ ($\Delta - .021$) | $.539_{\pm.015}$ |
| ROBERTA | $C\text{-}ENS_{high}$ | .745 | .540 | .623 | .504 | .711 | .588 | .624 | .626 | $.605_{\pm.017}$ ($\Delta - .005$) | $.608_{\pm.015}$ |
| | $C\text{-}ENS_{weigh}$ | .726 | .584 | .646 | .512 | .662 | .575 | .619 | .623 | $.611_{\pm.008}$ ($\Delta + .001$) | $.615_{\pm.008}$ |

Table 7: Cross-domain performance of cluster-based models, trained on EPIC and tested on SemEval 2018 Task 3 test set. In brackets the delta between the cluster-based and the correspondent demographic-based ensemble model.

above all baselines (Section 4.3). In-domain, the mining-based models perform rather well, with two PLMs (DISTILBERT and ROBERTA) performing better than their demographic-based counterparts, and BERT obtaining very similar scores. Cross-domain (Table 7), the results of the mining-based models are comparable to the demographic-based counterparts. The confidence-based ensembles with ROBERTA still perform better than their demographic-based counterparts, while the same is not true for the other two PLMs. This experiment shows that, by mining annotator perspectives, we can model annotator opinions based on their actual choices and look at irony as a phenomenon that transversely connects individuals of different demographics.

## 7 Conclusion

We showed how computational models of irony and hate speech are more proficient when built on disaggregated data, i.e., explicitly modeling the disagreement between the annotators. One step further, our experiments validate the *perspectivist* approach of grouping annotators to better encode their shared background, either by using their metadata or by automatically grouping them with a clustering approach. The main novelty of the method described in this paper is the use of the model confidence as a computational tool to ensemble a set of predictions coming from different perspective-aware models. This intuition, originated by the work of Frenda et al. (2023b), was proved central in building more accurate predictive models for irony and

hate speech detection.

The experimental results presented in this paper show that our method produces more accurate classifications of irony. In the case of hate speech, the results were mixed. However, the shape of the available data used for training the models in the two tasks varies greatly, in particular with respect to the number of annotators — EPIC (irony) has 74 annotators while MHS has thousands, resulting in a much greater variety of individual perspectives beyond those captured in the annotator metadata. More impressively, we have shown how the method is equally valid (and in many cases performing better) when the perspectives are automatically extracted from the disaggregated annotations rather than explicitly provided (Section 6).

Our experiments focused on the two case studies of irony and hate speech detection, and it remains to be explored how well our approach generalizes to other pragmatic phenomena. We plan to apply our methods to other pragmatic phenomena, irony-adjacent and in the undesirable language area, where we expect the most impact, as well as other languages. Future research should also include a broader range of demographic profiles to ensure greater inclusivity and improve the generalizability of perspectivist models across various linguistic contexts. Finally, perspective-aware models and ensembles naturally have the potential to improve explainability: they provide a layer of information that is useful for explaining the output of the classifiers, i.e., in terms of the human perspectives more likely to be sensitive to specific labels.

## Limitations

While the perspectivist approach for pragmatic phenomena classification shows promising results, there are certain limitations to consider. Firstly, our methodology relies on the availability of datasets data with disaggregated labels, showing different perspectives. This may not always be feasible or easily obtainable, especially for niche or under-resourced domains or languages.

The EPIC dataset has some limitations besides being monolingual. While efforts were made to ensure a fair balance in terms of the demographic profile of the annotators, the dataset is limited to five varieties of English tied to specific countries. This exclusion of other potential locations, such as New Zealand or Nigeria, as well as more nuanced distinctions among language varieties, may limit the dataset's representativeness and coverage of diverse linguistic perspectives.

About the self-identified gender dimension (provided by annotators themselves on the Prolific web application), we are aware of the wider spectrum of genders. However, this information was provided by the annotators only in a binary form.

Regarding the HS data, and in particular the MHS dataset, we aggregated annotations based on a selection of dimension influenced by factors like their distribution in the data. While it would have been unfeasible to train models on very sparse perspectives, we recognize that this may lead to their under-representation in our models.

## Ethics Statement

This research paper emphasizes the importance of incorporating diverse human perspectives in pragmatic phenomena classification, such as irony and hate speech detection, within Natural Language Processing tasks. We recognize the ethical considerations inherent in working with sensitive language data and promoting responsible research practices. The perspectivist approach in general, and this work in particular, aims at "giving voice to the few who hold a minority view" (Cabitza et al., 2023). Applied to the creation of language resources and the creation of automatic models, this principle leads to resources and models where bias is a controlled factor rather than undesirable criticality.

To ensure ethical conduct, we adopted measures to protect the privacy of annotators, and our data handling protocols are designed to safeguard personal information. We acknowledge the potential biases and ethical implications that may arise from the annotation process and strive to address them through careful analysis, discussion, and mitigation strategies. Furthermore, we advocate for the responsible use and deployment of perspectivist models in real-world applications. We emphasize the importance of fairness and inclusivity in model training and evaluation, aiming to minimize the amplification of biases or controversial viewpoints present in the annotations.

## Acknowledgements

The work of A. T. Cignarella, V. Patti, and C. Bosco was partially funded by the International project STERHEOTYPES - Studying European Racial Hoaxes and sterEOTYPES, funded by the Compagnia di San Paolo and VolksWagen Stiftung under the 'Challenges for Europe' Call for Projects (CUP: B99C20000640007).

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

# A Appendix

Table 8 shows the details of the different subsets of data extracted from EPIC to train the perspectivist models. The models were used to build the ensemble model described in Section 3 and the I-ENS and A-ENS baseline models presented in Section 4.3.

| Dimension | Value | # Instances | Socio-demographics |
|---|---|---|---|
| Gender | Female | 1,952 | Self-identified as female. |
| | Male | 2,023 | Self-identified as male. |
| Age | Boomer | 441 | Older than 58. |
| | GenX | 1,757 | Older than 42 and younger than 57. |
| | GenY | 1,964 | Older than 26 and younger than 41. |
| | GenZ | 1,124 | Younger than 25. |
| Nationality | UK | 1,365 | With English nationality. |
| | Indian | 1,175 | With Indian nationality. |
| | Irish | 1,296 | With Irish nationality. |
| | US | 1,352 | With American nationality. |
| | Australian | 1,377 | With Australian nationality. |

Table 8: Sub-sets of data extracted from EPIC on the basis of the socio-demographic traits of annotators.

Table 9 shows the details of the different subsets of data extracted from MHS, user in Section 5.

| Dimension | Value | Train | Val. | Test |
|---|---|---|---|---|
| Education | high | 24,021 | 2,971 | 2,956 |
| | low | 18,455 | 2,236 | 2,308 |
| Gender | female | 23,159 | 2,861 | 2,912 |
| | male | 20,072 | 2,517 | 2,465 |
| Ideology | conservative | 13,536 | 1,612 | 1,655 |
| | liberal | 19,376 | 2,433 | 2,416 |
| | neutral | 10,380 | 1,331 | 1,300 |
| Income | high | 22,489 | 2,755 | 2,762 |
| | low | 20,921 | 2,619 | 2,635 |
| Gold | | 28,387 | 3,498 | 3,503 |

Table 9: Sub-sets of data extracted from MHS on the basis of the socio-demographic traits of annotators.