# OpenReview forum: "Confidence-based Ensembling of Perspective-aware Models"
_EMNLP/2023/Conference — EMNLP 2023 Main_

### Official Review · Reviewer_ZE2f · 2023-07-23

**Soundness:** 4

**Excitement:**

3: Ambivalent: It has merits (e.g., it reports state-of-the-art results, the idea is nice), but there are key weaknesses (e.g., it describes incremental work), and it can significantly benefit from another round of revision. However, I won't object to accepting it if my co-reviewers champion it.

**Paper Topic And Main Contributions:**

In this paper, the authors propose a way to account for the annotators' uncertainty due to different demographic features. They train several classification models on data grouped by the annotators' demographics. In addition, they also propose a way to cluster the data if no annotators' demographic data is available.

**Questions For The Authors:**

1. General question, but let's look at Table 4 specifically. The difference between C-ENS and M-ENS in accuracy is in the third number ( 0.528 and 0.527 respectively). What is your uncertainty for the computations? If you round until the first or the second number, then you will get 0.5 and 0.53 in both cases - not better than the baseline, but way more effort. So, how is that justified the comparison in the third digit?
2. Why do you think you have the difference just in the third digit?
2a  do you plan to somehow address this issue for future work?
3. why did you choose three models from the same ''family''? you could have taken the strongest one
4.  Will the model for the data clustering be available and ready to use?

**Reasons To Accept:**

I like the idea of accounting for the annotator's demographics. There are multiple works urging us to take this path, and this work shows another dimension of how that could be incorporated into real research. In addition, the authors propose a method to get the annotator's data clustered even without demographic data, which might be useful for other researchers.
Another contribution is that authors train classifiers on data clustered by the annotator's demographic data, they conduct experiments on three models and in-domain and cross-domain experiments as well.

**Reasons To Reject:**

See questions

**Reproducibility:**

4: Could mostly reproduce the results, but there may be some variation because of sample variance or minor variations in their interpretation of the protocol or method.

**Reviewer Confidence:**

4: Quite sure. I tried to check the important points carefully. It's unlikely, though conceivable, that I missed something that should affect my ratings.

---

> ### Author Rebuttal · Authors · 2023-08-29
>
> We would like to thank the reviewer for the encouraging comments, and questions, to which we answer point by point:
>
> 1, 2: We calculated the standard deviation, which is at most 0.06 and about 0.02 on average or the macro-average F1-score and accuracy, indicating that the runs are very stable.
>
> 3: We aimed at testing the method with well established baselines, and therefore we selected two between the most widely used PLMs (BERT and RoBERTa), also in order to check the robustness of the method against different architectures. Since our method implies multiple fine tunings, with potential ecological and computational impact, we considered it important to further check its validity on a smaller model (DistilBERT).
>
> 4: We will make the code used to cluster annotators available, in order to reproduce the full experiment.
>
> On the topic of **reproducibility**, we are slightly surprised by the low score. All the dataset used in our study are publicly available, the main method and the baselines are documented in detail, and the source code to reproduce the results was attached to the submission as supplementary material and is in a Github repository that will be made public upon acceptance.

---

### Official Review · Reviewer_cKGw · 2023-08-02

**Soundness:** 4

**Excitement:**

4: Strong: This paper deepens the understanding of some phenomenon or lowers the barriers to an existing research direction.

**Paper Topic And Main Contributions:**

The work described in this paper focuses on building perspective-aware models of tasks whose annotation is highly subjective, such as irony and hate speech.  The perspective-aware models described here use the confidence scores produced by several models trained on different demographic subsets of the annotators of each data set to produce an overall confidence score and aggregate prediction.

**Questions For The Authors:**

Can you elaborate on why you chose distilBERT, BERT, and RoBERTa PLMs for fine-tuning?  Was the idea just to experiment with small, medium, and large (or larger) PLMs?

**Reasons To Accept:**

The paper is well-written and will motivate researchers to think about multiple annotation and how to use it to best represent different perspectives of different populations represented within that annotation.  To that end, it will get researchers thinking about conducting massive annotation efforts _with_ as many different populations represented as possible.

**Reasons To Reject:**

Some reviewers may not find this work particularly novel.

**Reproducibility:**

4: Could mostly reproduce the results, but there may be some variation because of sample variance or minor variations in their interpretation of the protocol or method.

**Reviewer Confidence:**

4: Quite sure. I tried to check the important points carefully. It's unlikely, though conceivable, that I missed something that should affect my ratings.

**Typos Grammar Style And Presentation Improvements:**

On line 011, "the key" should just be "key".

On line 019, "the classification" should just be "classification".

On line 083, this line would flow better as "on the one hand, we test whether...".

On line 107, "the classification" should just be "classification".

On line 109, "the attention" should just be "attention".

On line 124, "raise" should be "arise".

On line 125, "Whereas, data" should be "Data" because this sentence doesn't have a clause that supports the "whereas".

On line 164, insert "the" before "English".

On line 388, remove the "to" before "include".

On line 439, "resulted" should be "resulting".

---

> ### Author Rebuttal · Authors · 2023-08-29
>
> Thank you for the kind words and recognition of the contribution of our paper.
>
> Replying to the specific question, we aimed at testing the method with well established baselines, and therefore we selected two between the most widely used PLMs (BERT and RoBERTa), also in order to check the robustness of the method against different architectures. Since our method implies multiple fine tunings, with potential ecological and computational impact, we considered it important to further check its validity on a smaller model (DistilBERT).

---

### Official Review · Reviewer_17tq · 2023-08-05

**Typos Grammar Style And Presentation Improvements:** Line 388
**Soundness:** 3

**Excitement:**

2: Mediocre: This paper makes marginal contributions (vs non-contemporaneous work), so I would rather not see it in the conference.

**Paper Topic And Main Contributions:**

The paper proposes an approach to split the annotators based on their demographic values, assumes each of these values represent a perspective. It aggregates  perspective-aware models predictions while considering the models’ confidence.


**Questions For The Authors:**

A. Line 197, Section 4.2: What is the rationale for grouping each demographic and calling it a perspective? How can we determine if a specific demographic dimension forms a valid perspective? Assuming it is indeed a perspective, does the main contribution lie in the approach to combining these perspectives, while also claiming the importance of considering all of them?
B.What serves as the ground truth in this context? For each of the 11 splits, was the ground truth for training derived from the majority vote of that specific split? And during testing, were the 11 splits aggregated, and then compared to the overall majority vote?
C.The 11 splits are based on demographics. If a single text is annotated by multiple people within a split, how does the model learn from these annotations?
D.Line 378: What criteria were used to select only those four dimensions?
E.Are the observed performance differences statistically significant? It would be helpful to report the averages and standard deviations over repeated runs, as some reported values show less than a 0.01 difference, which might not be statistically significant.


**Reasons To Accept:**

Capturing the subjective viewpoints of annotators is a vital area of study, and the attempt to model these perspectives is of crucial importance.






**Reasons To Reject:**

Main issues with the proposed method:

If the model selects a label from just one perspective, it effectively disregards all the other ten perspectives. In such cases, the confidence corresponds to the model's confidence in its prediction. However, it is possible that the model's confidence is low because it didn't perfectly learn the perspective of a particular group, resulting in that label never being selected. This can be because of not correct splitting of perspective which is such a strong assumption to assumer that each demographic values share a perspective.
This approach contradicts the authors' claim that all perspectives need to be taken into account.
The authors acknowledged in Section 6 (line 471) that splitting based on demographics might not be representative of the perspective. However, a significant portion of the work relies on this approach, and conclusions are drawn based on assumptions that the authors are aware of being potentially flawed.
There are two research questions in this study, but they appear to be unrelated. One question focuses on considering confidence, while the other aims to extract perspective from annotations when metadata is not available.
The confidence score is almost identical to getting the soft labels, which has been proposed before
The suggested method involves aggregating the predicted labels, going beyond the majority vote of the models, and considering the soft labels. However, it remains unclear how this approach helps to incorporate more perspectives. The low confidence of a model could stem from various reasons, one of which may be the chosen and defined perspective. Annotations within a group might not necessarily share the exact same perspective merely due to a common demographic, and this could potentially lead to challenges in obtaining accurate soft labels.


**Reproducibility:**

4: Could mostly reproduce the results, but there may be some variation because of sample variance or minor variations in their interpretation of the protocol or method.

**Reviewer Confidence:**

4: Quite sure. I tried to check the important points carefully. It's unlikely, though conceivable, that I missed something that should affect my ratings.

---

> ### Author Rebuttal · Authors · 2023-08-29
>
> We would like to thank the reviewer for the many insights and useful comments, to which we reply by topic below.
>
> ### Splitting of perspectives
> We agree that demographics may not always align optimally with annotators perspectives, as discussed in recent works by [Santy et al.](https://aclanthology.org/2023.acl-long.505/) and [Orlikowsi et al.](https://aclanthology.org/2023.acl-short.88/) with mixed conclusions.
> In the ACL 2023 paper introducing the [EPIC dataset](https://aclanthology.org/2023.acl-long.774.pdf), it is shown that the demographic groups exhibit a higher inter-group agreement compared to the agreement of the whole set of annotations. The same holds for the [MHS dataset](https://aclanthology.org/2022.nlperspectives-1.11.pdf), for which Krippendorff’s alpha indicates that target groups (as defined by demographic data) exhibit higher inter-group agreement.
> The analysis of perspectivist datasets such as BREXIT (Akhtar et al. 2019, [2020](https://ojs.aaai.org/index.php/HCOMP/article/view/7473/7260)) comes to the same conclusion. Furthermore, while the clustering approach leads to similar results with our method (see Table 6), dividing the annotators based on demographics often has an advantage (Table 1), confirming the point that demographic factors strongly influence the perspectives of the annotators on irony in language.
>
> ### Ensembling
> Indeed, the ensemble classifiers presented in our work outputs a single prediction. However, this is strongly informed by the perspectivist learning made possible by a disaggregated dataset. Moreover, the comparison of perspectives-based ensembles with the two baselines (I-ENS and A-ENS) shows that the improvement of our model is motivated by taking into account annotator perspectives.
> One reason for proceeding this way is for the sake of evaluation, in- and cross-dataset. While a fully perspectivist evaluation framework has not been established yet, we point out in the future work section how we intend to tackle this relevant issue.
>
> ### Research questions
> The main result of our paper shows that model confidence is a key information to create perspectivist modeling. The section on using clustering techniques to extract the perspectives is an addition to the main line of research, that experimentally shows that, even when demographic information is not used, model confidence can still be used to create valid perspectivist models.
>
> ### Answers to the questions
> A: See above “splitting of perspectives”
>
> B, C: At training time, for each split, the ground truth is aggregated by majority vote of that specific split. To create the test sets, the majority vote is computed on all labels together.
>
> D. In the MHS corpus, those 4 dimensions are the ones annotated with categorical values.
>
> E. The result tables report the averages of 10 runs. We calculated the standard deviation, which is at most 0.06 and about 0.02 on average for the macro-average F1-score and accuracy, indicating that the runs are very stable. Thank you for this important suggestion, we will include all the figures in the final version.

---

### Meta-Review · Area_Chair_FVSb · 2023-09-17

**Recommendation:** 4

**Metareview:**

This paper introduces a method that accounts for annotators disagreement by grouping data based on demographic information about the annotators, for tasks where annotation is subjective (irony and hate speech detection).  The classifier aggreates confidence scores and predictions from models trained on annotation by annotators from different demographic groups.

This paper contributes to the growing literature on accounting for annotators disagreement. The claims are overall well-supported by the experiments, however, the presentation could be improved by clarifying what is meant by perspective and how that relates to demographic groups (reflecting for instance some of the points made in response to reviewer 17tq).

---

### Decision · Program_Chairs · 2023-10-07

**Decision:**

Accept-Main

**Comment:**

This paper introduces a method that accounts for annotators disagreement by grouping data based on demographic information about the annotators, for tasks where annotation is subjective (irony and hate speech detection).  The classifier aggreates confidence scores and predictions from models trained on annotation by annotators from different demographic groups.

This paper contributes to the growing literature on accounting for annotators disagreement. The claims are overall well-supported by the experiments, however, the presentation could be improved by clarifying what is meant by perspective and how that relates to demographic groups (reflecting for instance some of the points made in response to reviewer 17tq).